# A20 critically controls microglia activation and inhibits inflammasome-dependent neuroinflammation

Sofie Voet[1,2], Conor Mc Guire[1,2,3,4], Nora Hagemeyer[5], Arne Martens[1,2], Anna Schroeder[6,7], Peter Wieghofer[5,14], Carmen Daems[8], Ori Staszewski[5], Lieselotte Vande Walle[1,9], Marta Joana Costa Jordao [5], Mozes Sze[1,2], Hanna-Kaisa Vikkula[1,2], Delphine Demeestere[1,2], Griet Van Imschoot[1,2], Charlotte L. Scott[1,2], Esther Hoste [1,2], Amanda Gonçalves[1,2,10], Martin Guilliams [1,2], Saskia Lippens[1,2,10], Claude Libert[1,2], Roos E. Vandenbroucke [1,2], Ki-Wook Kim[11,15], Steffen Jung [11], Zsuzsanna Callaerts-Vegh[12], Patrick Callaerts[8], Joris de Wit[6,7], Mohamed Lamkanfi [1,9], Marco Prinz[5,13] & Geert van Loo[1,2]

Microglia, the mononuclear phagocytes of the central nervous system (CNS), are important for the maintenance of CNS homeostasis, but also critically contribute to CNS pathology. Here we demonstrate that the nuclear factor kappa B (NF-κB) regulatory protein A20 is crucial in regulating microglia activation during CNS homeostasis and pathology. In mice, deletion of A20 in microglia increases microglial cell number and affects microglial regulation of neuronal synaptic function. Administration of a sublethal dose of lipopolysaccharide induces massive microglia activation, neuroinflammation, and lethality in mice with microglia-confined A20 deficiency. Microglia A20 deficiency also exacerbates multiple sclerosis (MS)-like disease, due to hyperactivation of the Nlrp3 inflammasome leading to enhanced inter-leukin-1β secretion and CNS inflammation. Finally, we confirm a Nlrp3 inflammasome signature and IL-1β expression in brain and cerebrospinal fluid from MS patients. Collectively, these data reveal a critical role for A20 in the control of microglia activation and neuroinflammation.

[1] VIB Center for Inflammation Research, B-9052 Ghent, Belgium. [2] Department of Biomedical Molecular Biology, Ghent University, B-9052 Ghent, Belgium. [3] VIB Center for Medical Biotechnology, B-9052 Ghent, Belgium. [4] Department of Biochemistry and Microbiology, Ghent University, B-9052 Ghent, Belgium. [5] Institute of Neuropathology, Faculty of Medicine, University of Freiburg, D-79106 Freiburg, Germany. [6] VIB Center for Brain & Disease Research, B-3000 Leuven, Belgium. [7] Department of Neurosciences, KU Leuven, B-3000 Leuven, Belgium. [8] Department of Human Genetics, KU Leuven, B-3000 Leuven, Belgium. [9] Department of Internal Medicine, Ghent University, B-9052 Ghent, Belgium. [10] VIB Bio-Imaging Core, B-9052 Ghent, Belgium. [11] Department of Immunology, Weizmann Institute of Science, I-76100 Rehovot, Israel. [12] Laboratory of Biological Psychology, KU Leuven, B-3000 Leuven, Belgium. [13] BIOSS Centre for Biological Signalling Studies, University of Freiburg, D79106 Freiburg, Germany. [14] Institute of Anatomy, University of Leipzig, Leipzig D-04103, Germany. [15]Present address: Department of Pathology and Immunology, Washington University of Medicine, St. Louis, MO 63110, USA. These authors contributed equally: Sofie Voet, Conor Mc Guire, Nora Hagemeyer. These authors jointly supervised this work: Marco Prinz, Geert van Loo. Correspondence and requests for materials should be addressed to G. van L. (email: geert.vanloo@irc.vib-ugent.be)

The central nervous system (CNS) is considered an immune-privileged site where local immune defense is secured by a group of CNS-resident myeloid cells, including parenchymal microglia, macrophages at CNS interfaces, and a small number of dendritic cells. Microglia are distributed throughout the brain and spinal cord and are constantly surveilling the CNS to detect signs of pathogenic invasion or tissue damage. Upon activation, they orchestrate an innate immune response and are involved in clearing debris to restore CNS homeostasis[1]. Besides their role as immune sentinels, microglia also support and monitor synaptic function, control synaptogenesis and are vital for the survival of neurons during development[1,2]. However, it is increasingly recognized that under pathological conditions microglia can acquire a detrimental pro-inflammatory phenotype that actively contributes to the chronicity of inflammatory brain diseases[1]. These activated microglia not only secrete inflammatory cytokines and chemokines and present antigens to immune cells inducing CNS inflammation[1,3], they can also induce highly reactive A1 astrocytes that secrete a soluble toxin that directly kills neurons and oligodendrocytes[4].

Although, ontogenetically distinct, microglia share many surface markers with blood-derived monocytes, precluding their discrimination from infiltrating inflammatory monocytes and the study of their specific roles in inflammatory responses. Recently, a microglia targeting strategy in mice exploiting the CX3 chemokine receptor 1 (CX3CR1) promoter has been developed allowing the efficient deletion of a specific gene in microglia and other CNS macrophages[5–7]. Here we took advantage of this selective targeting strategy to address the role of microglia-expressed A20 in CNS homeostasis and pathology. Although, defects in A20-dependent regulation of nuclear factor kappa B (NF-κB)-dependent gene expression contributes to a variety of inflammatory and autoimmune diseases[8], its role in microglia activation is largely unexplored. Using conditional microglia A20 knockout mice, we here characterized the function of A20 in microglia under both physiological and pathological conditions.

## Results

### Characterization of mice with A20 deficiency in microglia.

Transcriptome analysis of purified brain cell types of mouse cortex identifies the A20 encoding gene *Tnfaip3/A20* as highly expressed in microglia, in contrast to its expression in neurons, astrocytes, endothelial cells, pericytes, and oligodendrocytes in various maturation stages[9]. To investigate the importance of A20 for microglia development and function, we first examined the expression of *A20* in different microglial developmental stages, including yolk sac precursors (EMPs, A1, A2), embryonic microglia and adult microglia (Fig. 1a). Although, *A20* is only marginally expressed in early development, an increase in expression is seen in the later stages of embryonic development, and A20 expression is further enhanced in adult microglia. To assess its function in vivo, we crossed mice carrying a floxed *A20* allele (*A20^FL*)[10] to *Cx3Cr1^CreErt2* transgenic mice allowing Cre-mediated gene deletion following tamoxifen (TAM) treatment (Supplementary Fig. 1a). Targeting of microglia using this system is based on microglia longevity and capacity of self-renewal without any appreciable input from circulating blood cells[5,11,12]. *A20^FL Cx3Cr1^CreErt2* mice were injected with TAM inducing A20 deletion in all Cx3Cr1-expressing cells. However, due to their short half-life, *Cx3Cr1 Ert2Cre^+* expressing blood-derived monocytes are gradually replaced by their monocyte precursors harboring non-rearranged *A20* alleles (Supplementary Fig. 1b). A20 deletion in microglia was confirmed at the protein level by western blotting of fluorescence-activated cell sorted (FACS) microglia ex vivo (Fig. 1b) and of primary microglia cultured

in vitro (Fig. 1c). In contrast, no deletion could be observed in bone marrow-derived macrophages (BMDM) and peritoneal macrophages in vitro or FACS-sorted Kupffer cells, showing A20 expression 4 weeks after TAM treatment (Supplementary Fig. 1c–e). In addition, we took advantage of the previously reported Cx3cr1Ert2CreER: Rosa26-fl-STOP-fl-YFP reporter mouse line[5,13] to analyze the level of Cre-mediated recombination events in Cx3Cr1-expressing macrophages that reside in other organs. In line with our other observations, we observed low to negligible YFP expression in heart, lung, liver, and skin macrophages, whereas 80–90% of microglia contained high YFP levels that remained relatively stable over time (Supplementary Fig. 1f). Together, these results demonstrate that *A20^FL Cx3Cr1^CreErt2* mice efficiently and selectively delete A20 expression in microglia.

Mice with TAM-induced A20 deficiency in microglia (*A20^FL/FL Cx3Cr1^CreErt2*, hereafter named A20^Cx3Cr1-KO) do not display overt spontaneous abnormalities, and histological analysis of brain sections of A20^Cx3Cr1-KO mice do not reveal any gross morphological defects, astrogliosis or demyelination (Supplementary Fig. 2). However, immunohistochemical examination with the microglia marker Iba-1⁺ revealed a significant increase in the number of microglia both in brain ($p = 0.0286$) and spinal cord sections of A20^Cx3Cr1-KO mice compared with control littermates (Fig. 1d–f, Supplementary Fig. 3a). The enhanced presence of microglia in A20^Cx3Cr1-KO CNS was confirmed by manual counting of Iba-1⁺ microglia and by FACS analysis of CD45^int CD11b⁺ cells from brain, implicating increased microglial proliferation as demonstrated by Ki67 immunostaining one week post TAM injection (Supplementary Fig. 3b). Quantitative morphometric three-dimensional (3D) analysis of Iba-1⁺ microglia in the adult cortex showed that A20-deficient microglia have an altered morphology, characterized by significantly longer processes and increased number of segments, branching, and terminal points relative to microglia from control mice (Fig. 1g, h and Supplementary Fig. 4). Together, these data suggest that lack of A20 induces proliferation of adult microglia with altered morphology.

Next, to determine whether A20 deficiency in microglia affects microglia activation or its immune function under homeostatic conditions, we expression profiled sorted CD45^int CD11b⁺ microglia from brains of A20^Cx3Cr1-KO and control littermate mice by quantitative deep sequencing of RNA transcripts. We observed significant differences in the mRNA profiles of microglial genes in A20^Cx3Cr1-KO relative to control mice. In total, 216 genes were significantly upregulated and 39 genes were downregulated ($p < 0.01$ and fourfold change) in A20^Cx3Cr1-KO microglia compared with controls (Supplementary Fig. 5a and Supplementary Data 1). Many genes typically expressed under physiological conditions (*P2ry12, Cx3cr1, Sall1, Gpr34, Fcrls, Rhob, Olfml3*) are selectively downregulated Bijschrijven dat het significant verschillend is? in A20^Cx3Cr1-KO microglia (Fig. 1i and Supplementary Fig. 5b). Among all upregulated transcripts in A20^Cx3Cr1-KO microglia, we identified genes linked to cell activation (*Cd45, F4/80, Cd86, Cd40, Cd11c*), microglia polarization (*Cxcl10, C4a/C4b, Tnfsf10, Cxcl9, Ccl2, Ccl5, Tlr1, Il1b*), MHC class I (*H2k1, H2d1, B2m, H2q7, H2m3, H2q8*), type I interferon signaling (*Irf-1, Irf-7, Irf-9, Stat1, Stat2, Ifi35, Ifit1, Ifit3, IfitM3*), and inflammatory signaling (*Nfkb1, Nfkb2, Relb, Il1b, Il12b, Ccl2, Ccl5, Ccl12, Cxcl9, Cxcl10, Cxcl11, Cxcl13*) (Fig. 1j and Supplementary Fig. 5c). Next, we analyzed the expression of "disease-associated microglia" (DAM) markers that have recently been identified[14], and demonstrate the significant upregulation of many of these DAM genes (*Axl, Cst7, Ctsl, Cd9, Csf1, Itgax, Clec7a, Lilrb4, Timp2, Ctsd, Ctsb*) in A20^Cx3Cr1-KO microglia (Supplementary Fig. 5d). Also the gene encoding for apolipoprotein E (*Apoe*), which has also been linked to a microglia disease-associated phenotype[15], is significantly upregulated in

A20$^{Cx3Cr1-KO}$ microglia (Supplementary Fig. 5e). The biological roles of the genes that were differentially expressed between A20$^{Cx3Cr1-KO}$ and control microglia were examined using ingenuity pathway analysis (IPA). Network analysis on differentially expressed genes identified "antimicrobial response, inflammatory response, and cell signaling" as the most differential between both populations, and IPA canonical pathway analysis identified pathways associated with interferon signaling, pattern

recognition receptor signaling, complement system, and inflammatory signaling as being enriched in A20$^{Cx3Cr1-KO}$ microglia (Supplementary Fig. 5f, g and Supplementary Table 1). Overall, the gene expression analysis suggests that upon A20 deletion microglia decrease their homeostatic "surveilling" state expression profile, and acquire an inflammatory, disease-associated signature. Finally, in agreement with our recent findings that other CNS macrophages, viz. the non-parenchymal macrophages in the subdural

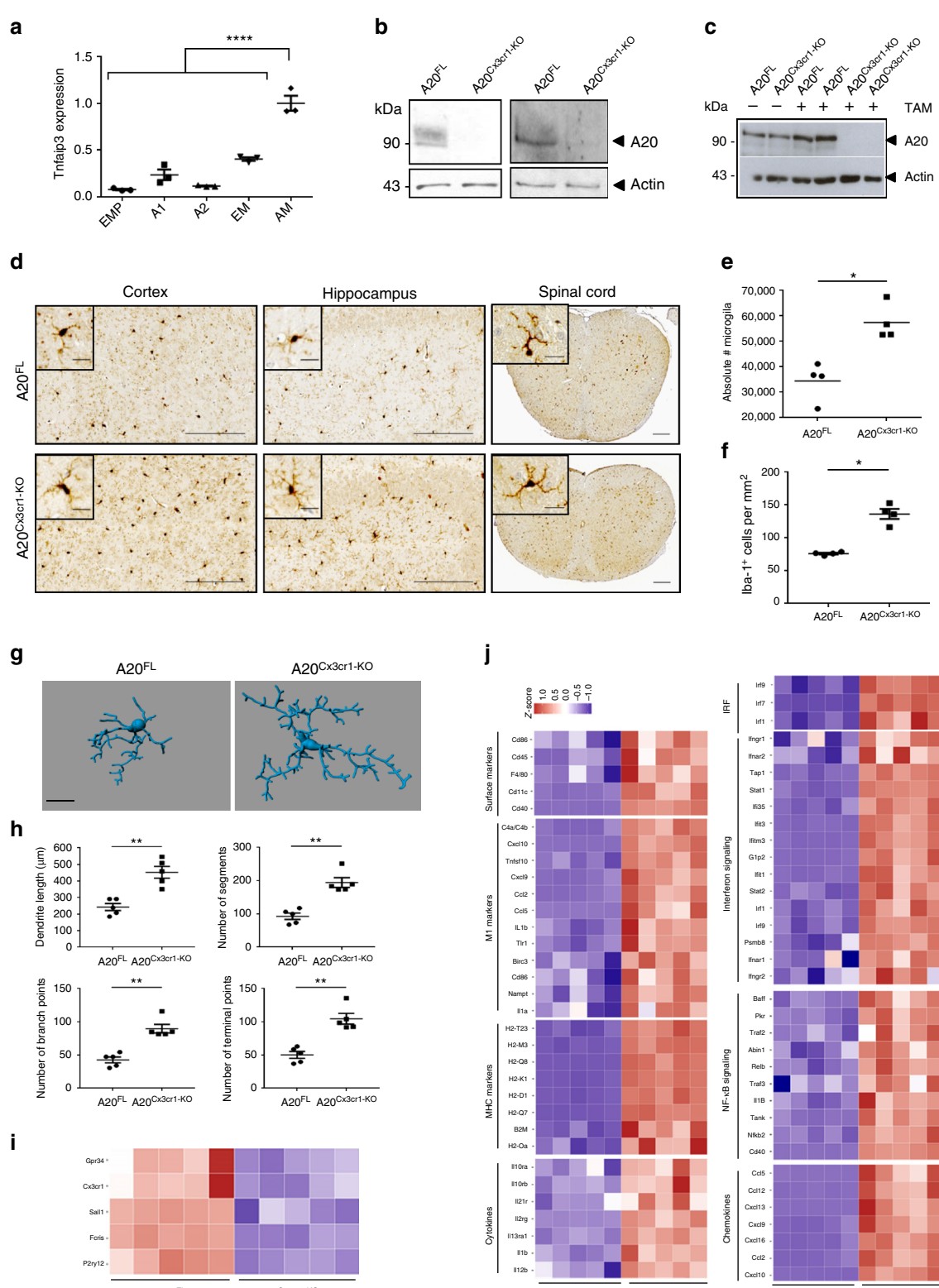

meningeal, perivascular spaces and choroid plexus, have the same origin and dynamics as microglia[7], we also identified, through RNA-seq analyses, significant differences in the mRNA profiles of A20-deficient CD45$^{hi}$CD11b$^+$CD206$^+$ macrophages relative to control CD45$^{hi}$CD11b$^+$CD206$^+$ macrophages. However, only 20 genes were significantly upregulated and 24 genes were down-regulated ($p < 0.01$ and fourfold change) in A20$^{Cx3Cr1-KO}$ CD45$^{hi}$CD11b$^+$CD206$^+$ macrophages compared with controls (Supplementary Fig. 6a and Supplementary Table 2). Also inflammatory genes are upregulated in these A20-deficient CD45$^{hi}$CD11b$^+$CD206$^+$ macrophages, although less pronounced as is the case for A20 deletion in microglia (Supplementary Fig. 6b). In conclusion, morphometric 3D analysis and genome signature comparison between microglia from A20$^{Cx3Cr1-KO}$ and control animals suggest a crucial role for A20 in the control of microglia activation under steady-state conditions, and demonstrate that in the absence of A20 microglia adopt a dysfunctional phenotype.

**A20 deficiency in microglia affects synaptic function.** Previous studies indicated that alterations in microglial number or activation can result in functional and structural deficits in cortical circuits[16,17]. To evaluate the functional consequences of increased microglial number and/or activation status for neuronal function, we prepared acute slices from P30–35 control and A20$^{Cx3Cr1-KO}$ mice, and analyzed spontaneous glutamatergic synaptic transmission in layer V pyramidal neurons in S1 cortex. No changes in AP spiking profiles or intrinsic membrane properties were found between pyramidal neurons from control and A20$^{Cx3Cr1-KO}$ mice (Fig. 2a, Supplementary Fig. 7, Table 1), indicating that cell-autonomous parameters are likely unaffected in A20$^{Cx3Cr1-KO}$ mice. However, we observed a marked increase in spontaneous excitatory postsynaptic current (sEPSC) frequency in A20$^{Cx3Cr1-KO}$ S1 layer V pyramidal neurons compared with those from control mice (Fig. 2b–e). The robust increase in sEPCS frequency in A20$^{Cx3Cr1-KO}$ mice with increased microglial number and (hyper)activation is consistent with previous studies showing an increased frequency of spontaneous synaptic transmission following microglia activation[18] and a decreased frequency following microglia depletion[16,17]. A small, but significant decrease in sEPSC amplitude was also observed (Fig. 2b, f, g), indicating that microglia activation could negatively influence the density of postsynaptic glutamate receptors on dendrites.

Next, to evaluate whether the observed aberrant excitatory synaptic function affects learning and memory, A20$^{Cx3Cr1-KO}$ and control littermate mice were examined in the hidden platform protocol of the Morris water maze (MWM)[19], a very reliable and robust protocol to investigate hippocampus-dependent spatial learning and memory[20]. Over ten training days, 6 month old female control and A20$^{Cx3Cr1-KO}$ mice showed similar learning curves to locate the hidden platform using distal visual cues. Interspersed probe trials on days 6 and 11 showed no difference in spatial reference memory and a robust preference for the correct target quadrant. Cognitive flexibility, a measure for executive function, was assessed in a consecutive 5 day reversal training where the platform position was relocated to the opposite quadrant. A20$^{Cx3Cr1-KO}$ and control mice showed similar learning curves (repeated measures two-way ANOVA for factor day: $F(4,76) = 16.8$; $p < 0.0001$) (Fig. 2h). Interestingly, during the reversal probe trial only control mice showed a significant preference for the target quadrant over the other three quadrants (repeated measures two-way ANOVA for factor quadrant preference: $F(3,57) = 11,4$; $p < 0.0001$) (Fig. 2i) and were at any given time point significantly closer to the location of the reversal platform than to the location of the acquisition platform (expressed as total distance to target) (two-way ANOVA for factor platform location: $F(1,38) = 14,36$; $p = 0.0005$) (Fig. 2j). Together these findings indicate that after 5 days of reversal learning, A20$^{Cx3Cr1-KO}$ mice do not have a robust spatial representation of the platform's correct location and were not capable of forming a robust reference memory. Together, these results show that the aberrantly activated microglia in A20$^{Cx3Cr1-KO}$ mice affect excitatory synaptic function with consequences for proper cognitive function.

**A20 in microglia protects against lipopolysaccharide (LPS)-induced inflammation.** Microglia are the CNS-resident immune cells acting as the first responders to microbial infections of the brain[21]. To investigate the importance of A20 for microglia function in a model of neuroinflammation, A20$^{Cx3Cr1-KO}$ and control littermate mice were injected with a single sublethal dose of the TLR4 agonist LPS, known to induce neuroinflammation through microglia activation and expression of pro-inflammatory cytokines[22,23]. Surprisingly, in contrast to control mice, which only exhibit a modest drop in body temperature in the first hours after systemic LPS injection, A20$^{Cx3Cr1-KO}$ mice displayed severe hypothermia (Fig. 3a) and increased mortality (Fig. 3b). Similarly, A20$^{Cx3Cr1-KO}$ mice presented severe hypothermia when LPS was administered by intracerebroventricular (icv) injection (Supplementary Fig. 8a). Histological analysis of the CNS 10 h after systemic LPS administration, a time point at which all A20$^{Cx3Cr1-KO}$ mice are still alive, demonstrated increased

**Fig. 1** CNS phenotype of A20$^{Cx3Cr1-KO}$ mice. **a** A20 expression during microglial cell development. Cells were sorted according to Kierdorf et al.[52]. A1 immature CD45$^+$ c-kit$^{lo}$ Cx3Cr1$^-$ cells (yolk sac), A2 mature CD45$^+$ c-kit$^-$ Cx3Cr1$^+$ cells (yolk sac), AM: adult microglia, EM: embryonic microglia (E14), EMP: erythromyeloid precursors (yolk sac). Each symbol represents one mouse, $n = 3$ per group. Data presented as mean ± SEM. Significant differences determined by a one-way ANOVA with Tukey correction for multiple comparison (****$p < 0.0001$). **b** Immunoblot for A20 expression on ex vivo FACS-sorted microglia from control (A20$^{FL}$) and A20$^{Cx3Cr1-KO}$ mice 4 (left) or 35 weeks (right) after TAM injection. Actin is shown as loading control. **c** Immunoblot for A20 expression on lysates from primary microglia from control (A20$^{FL}$) and A20$^{Cx3Cr1-KO}$ mice after stimulation with 4-OH-TAM. Actin is shown as loading control. Data are representative of two independent experiments. **d** Immunohistochemistry for Iba-1$^+$ in the cerebral cortex, hippocampus, and spinal cord of control (A20$^{FL}$) and A20$^{Cx3Cr1-KO}$ mice. Scale bars: 100 μm (cortex and hippocampus) and 200 μm (spinal cord), insert: 10 and 20 μm, respectively. Representative images are displayed. **e** Flow cytometric quantification of the number of CD45$^{int}$ CD11b$^+$ microglia in brain of control (A20$^{FL}$) and A20$^{Cx3Cr1-KO}$ mice 4 weeks post TAM injection. Each symbol represents one mouse. Data are representative of two independent experiments and presented as mean ± SEM. Significant differences determined by Mann–Whitney U-statistical test (*$p < 0.05$). **f** Number of Iba-1$^+$ ramified parenchymal microglia. Each symbol represents one mouse, with four mice per group. Data presented as mean ± SEM. Significant differences determined by a Mann–Whitney U-statistical test (*$p < 0.05$). **g, h** 3D reconstruction (scale bars: 10 μm) (**g**) and Imaris-based quantification of cell morphology (**h**) of cortical Iba-1$^+$ microglia. Each symbol represents one mouse; three cells analyzed per mouse; $n = 5$ per condition. Data presented as mean ± SEM. Significant differences determined by a Mann–Whitney U-statistical test (**$p < 0.01$). **i, j** RNA sequencing on FACS-sorted microglia from TAM-injected control (A20$^{FL}$) and A20$^{Cx3Cr1-KO}$ mice. Each column represents microglia from one individual mouse, $n = 5$ per group. Color code presents linear scale. Pathway analysis of RNA-seq datasets demonstrates downregulation of homeostatic genes (**i**) and upregulation of genes involved in microglia activation, polarization, MHC class I, interferon and inflammatory signaling (**j**) in A20$^{Cx3Cr1-KO}$ microglia

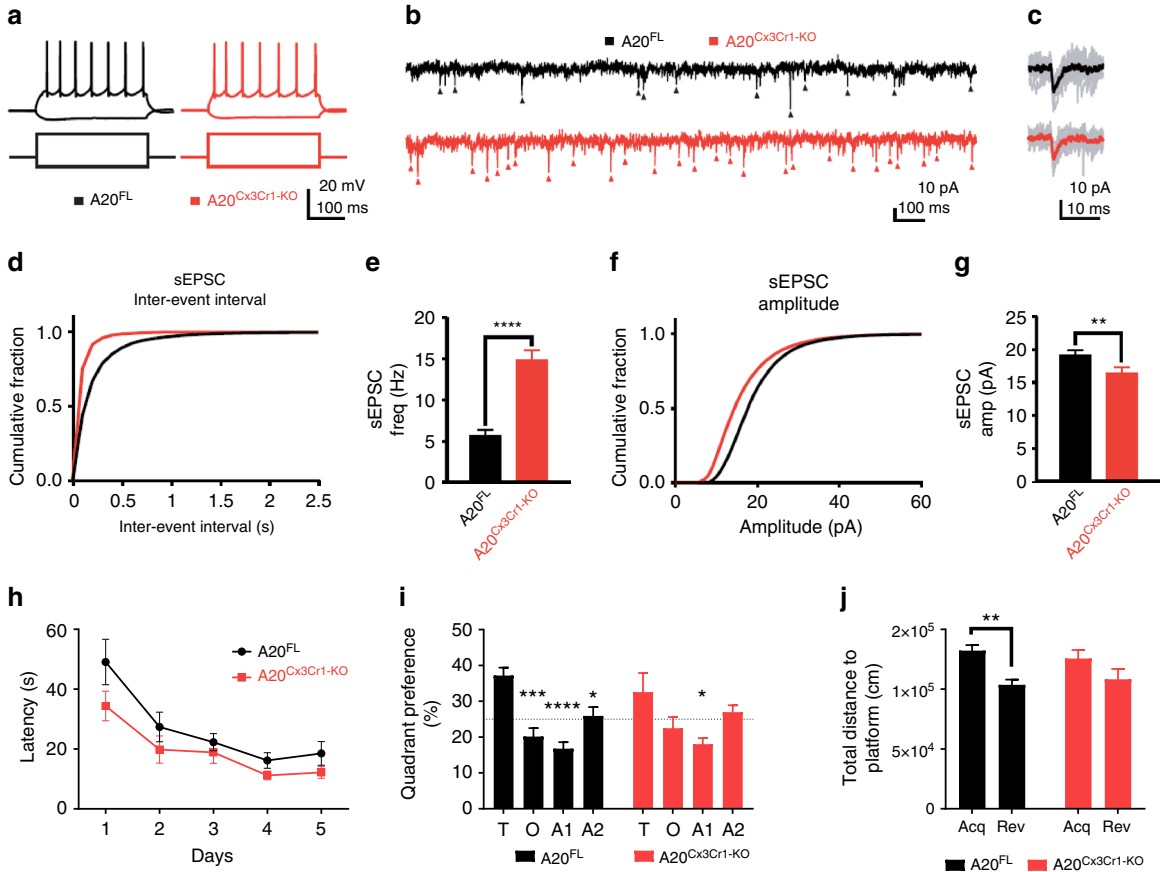

**Fig. 2** Effect of microglia hyperactivation on basal network activity in somatosensory pyramidal neurons. **a** Example action potential (AP) firing profiles, showing the typical regular spiking behavior of layer V S1 pyramidal neurons both in control and in A20$^{Cx3Cr1-KO}$ neurons. Responses to hyperpolarizing (−150 pA) and suprathreshold (400 pA) current injections are shown. **b** Example sEPSC traces, 2s stretches. **c** Examples of individual sEPSC events. Individual events shown in gray, averaged event shown as overlay (control, black; A20$^{Cx3Cr1-KO}$, red). **d** Cumulative probability distributions of inter-event sEPSC intervals (IEIs) for control (black) or A20$^{Cx3Cr1-KO}$ (red) cells. **e** sEPSC frequency (control (black) 5.76 ± 0.58 Hz, $n/m = 21/3$; A20$^{Cx3Cr1-KO}$ (red) 14.8 ± 1.29 Hz, $n/m = 20/3$; ****$p < 0.0001$). Data represents means ± SEM. Number of cells ($n$), number of animals ($m$). Statistical differences were determined by a two-tailed Wilcoxon matched-paired $t$-test. **f** Cumulative probability distributions for sEPSC amplitudes in control (black) or A20$^{Cx3Cr1-KO}$ (red) cells. **g** sEPSC amplitudes (control 19.3 ± 0.69 pA, $n/m = 21/3$; A20$^{Cx3Cr1-KO}$ 16.5 ± 0.87 pA, $n/m = 20/3$; **$p = 0.0013$). Data represents means ± SEM. Number cells ($n$), number of animals ($m$). Statistical differences were determined by a two-tailed unpaired Mann–Whitney $t$-test. **h** Cognitive flexibility in Morris water maze reversal learning was similar in control (A20$^{FL}$, $n = 12$) and A20$^{Cx3Cr1-KO}$ mice ($n = 9$). **i** In the reversal probe trial, controls displayed significant target quadrant preference, while A20$^{Cx3Cr1-KO}$ mice did not (T: target quadrant, O: opposite quadrant, A1: adjacent 1 quadrant, A2: adjacent 2 quadrant). Chance level at 25% is indicated. **j** During the reversal probe trial, control animals searched closer to the correct reversal location for the platform than to the old location (distance to platform), while A20$^{Cx3Cr1-KO}$ mice searched at equidistance to both locations. All data (**h–j**) are represented as means ± SEM and statistical differences were determined by repeated measures two-way ANOVA (**h, i**) and two-way ANOVA (**j**) using Bonferroni correction for post hoc analysis (**h–j**) (*$p < 0.05$, **$p < 0.01$, ***$p < 0.001$, ****$p < 0.0001$)

**Table 1 Intrinsic properties of pyramidal neurons from control (A20$^{FL}$) and A20$^{Cx3Cr1-KO}$ mice (NS, not significant)**

| | A20$^{FL}$ (mean ± SEM) | A20$^{Cx3Cr1-KO}$ (mean ± SEM) | $p$-value |
|---|---|---|---|
| *Passive properties* | | | |
| Input resistance, MΩ | 58.8 ± 4.198 | 60.7 ± 4.783 | 0.435 (NS) |
| Resting membrane potential, mV | −58.8 ± 1.393 | −59.8 ± 1.756 | 0.670 (NS) |
| Membrane capacitance, pF | 25.3 ± 0.649 | 26.45 ± 0.813 | 0.465 (NS) |
| Membrane time constant, ms | 9.2 ± 0.673 | 9.6 ± 0.651 | 0.461 (NS) |
| *Action potential (AP) features* | | | |
| First spike latency, ms | 87.5 ± 11.999 | 67.2 ± 11.873 | 0.310 (NS) |
| Interspike frequency, Hz | 16.4 ± 3.566 | 18.6 ± 3.786 | 0.360 (NS) |

microgliosis in A20$^{Cx3Cr1-KO}$ brain and spinal cord compared to control tissue (Fig. 3c). Since A20 knockout mice with A20-deficient microglia already spontaneously display a pro-inflammatory phenotype characterized by microglia

hyperproliferation (Fig. 1), no further effect on microglia proliferation is seen after LPS (Fig. 3d). Microglia morphology analysis 10 h post LPS suggests a reactive phenotype in A20$^{Cx3Cr1-KO}$ mice compared with wild-type littermates, as

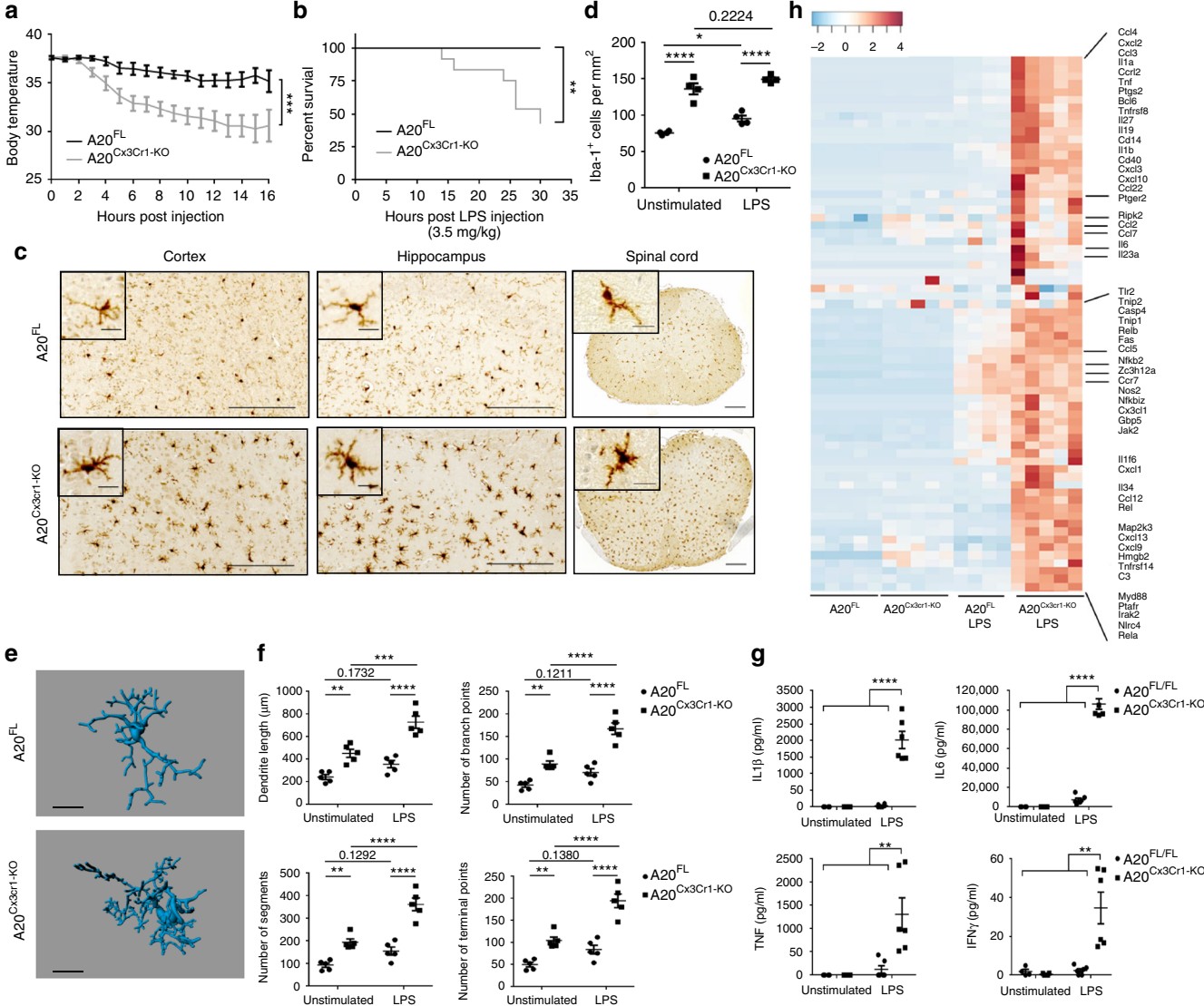

**Fig. 3** A20^Cx3Cr1-KO mice are hypersensitive to LPS. **a** Rectal body temperature responses and **b** survival were analyzed in function of time in control (A20^FL; n = 13) and A20^Cx3Cr1-KO (n = 14) mice after intraperitoneal injection of 3.5 mg/kg LPS. The combined results of three independent experiments are shown. Body temperature data are means ± SEM. Statistical differences were determined by a REML analysis for rectal body temperatures and a Mantel–Cox test for the survival curve (**p < 0.01, ***p < 0.001). **c** Immunohistochemistry images showing Iba-1+ expression in the cerebral cortex, hippocampus and spinal cord of control (A20^FL) and A20^Cx3Cr1-KO mice 10 h post LPS challenge. Scale bars represent 100 µm (cortex and hippocampus) and 200 µm (spinal cord), insert: 10 and 20 µm, respectively. Representative images are displayed. Data are representative of two independent experiments. **d** Number of Iba-1+ ramified parenchymal microglia in brain. Each symbol represents data from one mouse, n = 4 per group. Data are presented as mean ± SEM. Significant differences were determined by a two-way ANOVA with Tukey correction for multiple comparison (*p < 0.05, ****p < 0.0001). **e, f** Three-dimensional reconstruction (scale bars represent 10 µm, **e**) and Imaris-based semiautomatic quantification of cell morphology (**f**) of cortical Iba-1+ microglia. Each symbol represents the average of three measured cells per mouse; n = 5 per group. Data are presented as mean ± SEM. Significant differences were determined by two-way ANOVA with Tukey correction for multiple comparison (**p < 0.01, ***p < 0.001, ****p < 0.0001). **g** Bioplex analysis of cytokine levels in CSF 10 h post LPS. Each symbol represents one mouse. Data are representative of three independent experiments. Significant differences were determined by a one-way ANOVA with Tukey correction for multiple comparison (**p < 0.01, ****p < 0.0001). **h** RNA prepared from FACS-sorted microglia from control (A20^FL) and A20^Cx3Cr1 mice either or not injected with LPS for 10 h was submitted for RNA sequencing. Heat map of expression values for inflammatory genes that are upregulated in LPS-stimulated A20^Cx3Cr1 microglia compared with LPS-stimulated control microglia. Each column represents microglia data from one individual mouse, with four or five mice per group. Color code presents linear scale

evidenced by a significantly higher number of processes, branching points, terminal point, and segments. (Fig. 3e, f and Supplementary Fig. 4). In agreement, significantly elevated levels of inflammatory cytokines were detected in CNS tissue and cerebrospinal fluid (CSF) of A20^Cx3Cr1-KO relative to littermate A20^FL mice after systemic LPS challenge (Fig. 3g and Supplementary Fig. 8b–d). Finally, microglia from LPS-injected

A20^Cx3Cr1-KO and A20^FL control mice were isolated and global gene expression was determined by RNA sequencing. Investigation of the gene ontology enrichment network on differentially expressed genes revealed that inflammatory pathways are highly affected in LPS-stimulated microglia compared with unstimulated microglia, as expected. However, 2774 genes were differentially expressed between A20^Cx3Cr1-KO and control microglia isolated

10 h after systemic LPS injection (cut-off: $p < 0.01$ and at fold change $\geq 4$; 1609 upregulated, 1165 downregulated in A20$^{Cx3Cr1-KO}$ compared with control), and LPS-stimulated A20$^{Cx3Cr1-KO}$ microglia expressed much higher levels of inflammatory cytokine and chemokine pathways compared with LPS-stimulated control microglia, demonstrating their hyperactivation (Fig. 3h and Supplementary Fig. 9a–c). In agreement, the homeostatic gene signature of LPS-stimulated microglia is strongly and significantly suppressed compared with non-stimulated microglia (Supplementary Fig. 9d). Together, these results demonstrate that A20 expression in microglia controls inflammatory CNS responses and is essential to prevent a detrimental response to LPS-induced neuroinflammation.

**Nlrp3 inflammasome hyperactivation in A20-deficient microglia.** An inflammatory cytokine that was significantly upregulated in microglia from LPS-stimulated A20$^{Cx3Cr1-KO}$ mice and can be detected in the CSF of A20$^{Cx3Cr1-KO}$ mice was IL-1β (Fig. 3g, h). LPS induced a strong upregulation of pro-IL-1β in both control and A20$^{Cx3Cr1-KO}$ microglia compared with non-injected mice, however, this upregulation is significantly stronger in microglia from LPS-injected A20$^{Cx3Cr1-KO}$ mice compared with LPS-

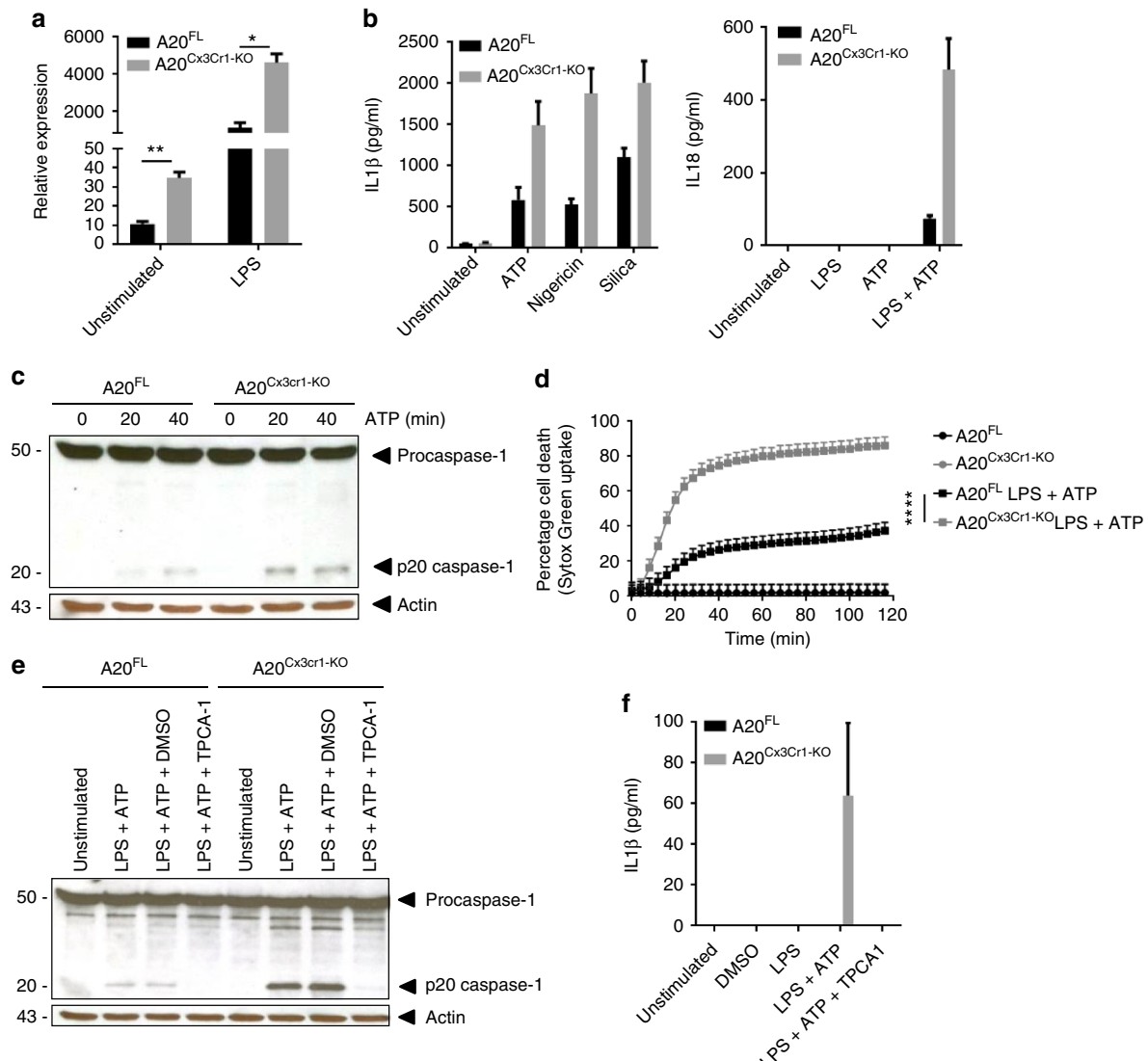

**Fig. 4** A20$^{Cx3Cr1-KO}$ microglia are hypersensitive to Nlrp3 inflammasome activation. **a** IL-1β mRNA expression in microglia from A20$^{Cx3Cr1-KO}$ mice compared with control mice either or not injected with LPS. Data are presented as mean ± SEM. Significant differences were determined by a Mann–Whitney $t$-test (*$p < 0.05$, **$p < 0.01$). **b** IL-1β and IL-18 protein levels in the supernatant of primary cultured microglia stimulated with LPS alone or together with ATP, silica, or nigericin. Data represent the mean ± SD of three technical replicates of pooled microglial cells from control (A20$^{FL}$) and A20$^{Cx3Cr1-KO}$ mice. Data are representative of three independent experiments. **c** Immunoblot for procaspase-1 and cleaved caspase-1 (p20) in primary cultured microglia from control (A20$^{FL}$) and A20$^{Cx3Cr1-KO}$ mice stimulated with LPS and/or ATP. Actin is shown as loading control. Data are representative of two independent experiments. **d** Pyroptosis induction in primary microglia from control (A20$^{FL}$) and A20$^{Cx3Cr1-KO}$ mice stimulated with LPS and ATP, as measured by Sytox Green uptake. Data are presented as mean ± SEM and are representative of two independent experiments. Significant differences were determined by a REML analysis (****$p < 0.0001$). **e** Immunoblot showing procaspase-1 and cleaved caspase-1 in primary cultured microglia from control (A20$^{FL}$) and A20$^{Cx3Cr1-KO}$ mice either or not pretreated in vitro with TPCA-1 and stimulated with LPS and/or ATP. Actin is shown as loading control. Data are representative of two independent experiments. **f** IL-1β protein in supernatant of primary cultured microglia from control (A20$^{FL}$) and A20$^{Cx3Cr1-KO}$ mice either or not pretreated in vitro with TPCA-1 and stimulated with LPS and/or ATP. Data represent the mean ± SD of three technical replicates of pooled microglial cells, and are representative of two independent experiments

injected control mice (Fig. 4a). IL-1β is the prototype cytokine secreted by cells upon activation of inflammasomes, multi-protein complexes that enable the activation of caspase-1 leading to the processing and secretion of biologically active IL-1β[24]. Also IL-18 is produced through the activation of inflammasomes[24], however, no difference in IL-18 expression could be observed between FACS-sorted microglia from LPS-injected A20$^{Cx3Cr1-KO}$ and control mice. Microglia are capable of engaging different inflammasome types in response to infectious agents and host-derived danger signals that are associated with neurological diseases[25]. To study the contribution of inflammasome signaling to inflammatory responses of A20$^{Cx3Cr1-KO}$ microglia, we assessed caspase-1 processing in primary cultured microglia isolated from A20$^{Cx3Cr1-KO}$ and control littermate mice. Secreted levels of IL-1β and IL-18 were significantly increased in LPS-primed A20$^{Cx3Cr1-KO}$ microglia that were treated with soluble (ATP and nigericin) or crystalline (silica) stimuli of the Nlrp3 inflammasome compared with control microglia (Fig. 4b). In accordance, caspase-1 autoprocessing was substantially increased in A20$^{Cx3Cr1-KO}$ relative to control microglia (Fig. 4c and Supplementary Fig. 10a). Similarly, caspase-1 processing and IL-1β/IL-18 secretion were also increased in A20$^{Cx3Cr1-KO}$ microglia primed with the TLR2 agonist Pam3CSK4 and treated with ATP (Supplementary Fig. 10b, c). Besides their roles in maturation and secretion of IL-1β and IL-18, a major effector mechanism of inflammasomes is the induction of pyroptosis, a pro-inflammatory and lytic mode of cell death occurring mainly in myeloid cells including microglia[25]. Pyroptosis induces cell swelling and rupture of the plasma membrane, causing massive leakage of cytosolic contents provoking inflammatory reactions. Similarly to the effect on caspase-1 autoprocessing and IL-1β/IL-18 secretion, the induction of cell death was also enhanced in A20$^{Cx3Cr1-KO}$ microglia (Fig. 4d and Supplementary Fig. 10d). Activation of the Nlrp3 inflammasome, one of the most pleiotropic inflammasomes, requires a priming signal that results in the upregulation of Nlrp3 expression along with the inflammasome substrate pro-IL-1β via the pro-inflammatory transcription factor NF-κB[26]. A20 negatively regulates Nlrp3 inflammasome signaling by suppressing NF-κB-dependent production of Nlrp3 and pro-IL-1β in macrophages[27]. Indeed, a pharmacological inhibitor of IKK2 (inhibitor of NF-κ-B2), TPCA-1, significantly reduced ATP-induced caspase-1 autoprocessing and IL-1β secretion in LPS-primed A20$^{Cx3Cr1-KO}$ microglia (Fig. 4e, f and Supplementary Fig. 10e). Together, these results demonstrate that A20 negatively controls Nlrp3 inflammasome priming and activation in microglia.

**Mice with A20-deficient microglia are hypersensitive to experimental autoimmune encephalomyelitis (EAE).** Based on our results, we investigated the microglia-specific function of A20 in a model of autoimmune CNS inflammation, namely EAE. A20$^{Cx3Cr1-KO}$ and control mice were immunized with a myelin oligodendrocyte glycoprotein (MOG) peptide (MOG$_{35-55}$) and disease progression was monitored by assessing clinical disease symptoms and body weight (Fig. 5a, b). Both A20$^{Cx3Cr1-KO}$ and control mice developed EAE, however, A20$^{Cx3Cr1-KO}$ mice developed earlier disease onset and exhibited a more severe disease course as compared with control mice (Fig. 5a, b and Table 2), demonstrating a crucial role for microglial A20 activity in EAE pathogenesis. In contrast, A20 deletion in other CNS cell types (all CNS progenitor cells, neurons, astrocytes, or oligodendrocytes) did not result in differences in EAE clinical pathology (Supplementary Fig. 11a). Clinical pathology in A20$^{Cx3Cr1-KO}$ was confirmed by histology and flow cytometry on spinal cord sections at start of the clinical manifestations, showing extensive demyelination, axonal damage, inflammation and immune cell infiltration in A20$^{Cx3Cr1-KO}$ mice, while nearly no immune cell infiltration, demyelination, or axonal loss could be detected in the spinal cord of control mice at this early time point (Fig. 5c, d, Supplementary Fig. 11b). Also the expression of inflammatory cytokines, chemokines and TH1-, TH17-, and Treg-linked factors, confirmed the hypersensitivity of A20$^{Cx3Cr1-KO}$ mice to EAE (Supplementary Fig. 11c). Although, the Cx3Cr1 promoter is not thought to target lymphocytes in $Cx3Cr1^{CreErt2}$ mice, we confirmed that the enhanced sensitivity in EAE was caused by the CNS-specific ablation of A20 and not by an impaired peripheral T-cell response. Lymphocytes from immunized mice were isolated and tested in vitro for their response upon secondary exposure to MOG$_{35-55}$ peptide, showing similar responses in A20$^{Cx3Cr1-KO}$ and control mice, thereby demonstrating that peripheral immune functions are not affected in A20$^{Cx3Cr1-KO}$ mice (Supplementary Fig. 11d). As A20 deficiency enhances the activation of the Nlrp3 inflammasome in isolated primary microglia (Fig. 4), we speculated that the hyperactivation of the Nlrp3 inflammasome is responsible for the aggravated EAE phenotype in A20$^{Cx3Cr1-KO}$ mice. Indeed, gene expression analysis of spinal cord tissue showed enhanced expression of pro-IL-1β, ASC, Nlrp3, and caspase-1 in A20$^{Cx3Cr1-KO}$ mice compared with control mice, suggesting a contribution of inflammasome activities in the observed phenotype (Fig. 5e). To test this hypothesis, Nlrp3$^{-/-}$ mice were crossed with A20$^{Cx3Cr1-KO}$ mice and EAE disease development was monitored. In contrast to A20$^{Cx3Cr1-KO}$ mice, which develop a more severe pathology, A20$^{Cx3Cr1-KO}$-Nlrp3$^{-/-}$ mice show a significantly reduced clinical pathology, comparable to control A20$^{FL}$ littermate mice (Fig. 5f). These results are in line with previous studies demonstrating a critical role for Nlrp3 in the development of EAE by mediating peripheral immune responses[28–30]. However, Nlrp3$^{-/-}$ mice lack the Nlrp3 protein ubiquitously, and the specific contribution of inflammasome signaling inside the CNS to EAE pathogenesis is unknown. To study the specific function of inflammasomes within microglia during EAE, caspase-1 conditional knockout mice having a floxed allele (caspase-1$^{FL}$) were generated[31] (Supplementary Fig. 12a) and crossed with A20$^{Cx3Cr1-KO}$ mice to produce mice lacking both A20 and caspase-1 in microglia (caspase-1-A20$^{Cx3Cr1-KO}$). In contrast to single A20$^{Cx3Cr1-KO}$ mice, caspase-1/A20$^{Cx3Cr1-KO}$ mice developed less severe disease upon immunization with MOG$_{35-55}$ peptide (Fig. 5g). However, caspase-1-A20 double deficient microglia still display an inflammatory phenotype, similar to A20 knockout microglia (Supplementary Fig. 12b), suggesting that downstream effectors other than caspase-1 contribute to inflammatory gene upregulation. Together, these results demonstrate the role of A20 in the control of Nlrp3 inflammasome activation locally in microglia, and indicate the importance of this microglial A20/Nlrp3 inflammasome axis in EAE pathogenesis.

Finally, to investigate the relevance of our findings to human CNS pathology, postmortem brain tissue and CSF of multiple sclerosis (MS) patients (Supplementary Table 3) were analyzed for expression of A20 and for the presence of an inflammasome "signature." Expression of A20/TNFAIP3, as well as the expression levels of IL-1β and NLRP3 were significantly increased in MS plaques compared with normal appearing white matter, and a trend, albeit not significant, in enhanced IL-18 and caspase-1 in MS plaques could be observed (Fig. 6a, b). Higher IL-18 and IL-1β protein levels were also detected in CSF of MS patients compared with controls, suggestive of inflammasome activation in microglia, although also other immune cells may have contributed to this (Fig. 6c). Our data analyzing the expression levels of inflammasome mediators in brain plaques isolated from normal and MS patients show that brain tissue affected by MS

clearly exhibits enhanced activation of the NLRP3 inflammasome, as shown in previous studies.

## Discussion

In this study, we identified A20 as a crucial mediator of microglia activation, synaptic function, and EAE pathogenesis. In contrast to its expression in neurons, astrocytes, oligodendrocytes (both precursor and mature myelinating), and brain endothelial cells, A20 is highly expressed in adult microglia[9]. Mice lacking A20 specifically in microglia have increased total numbers of microglia, in the absence of any overt abnormalities. Detailed morphometric analysis, as well as transcriptome analysis of

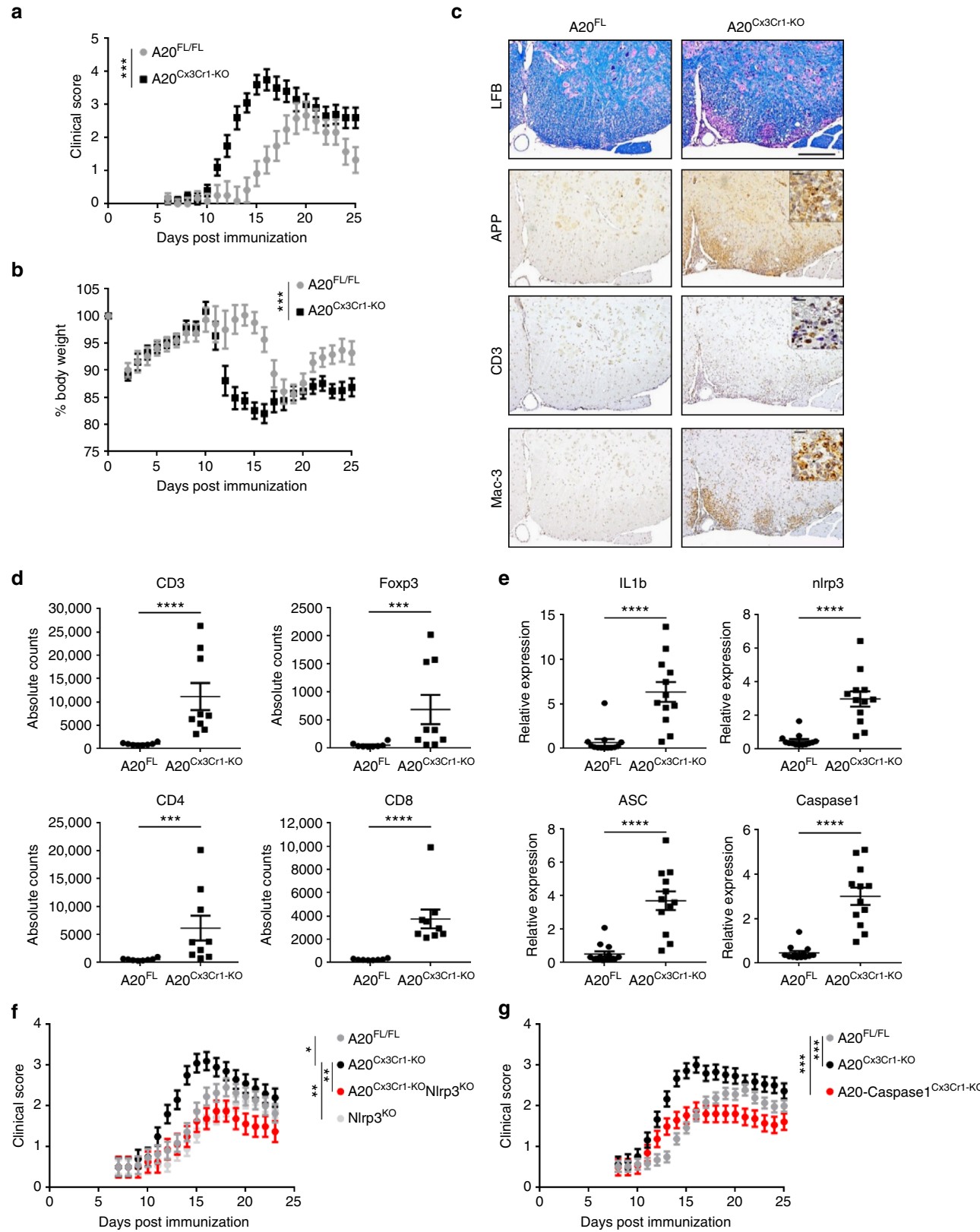

microglia from A20-deficient mice revealed an altered microglia phenotype. In agreement, our expression study revealed an inflammatory "signature" of these steady-state A20-deficient microglia, demonstrated by the upregulation of many inflammatory genes and pathways. Also many DAM markers, including apolipoprotein A (*Apoe*), are significantly upregulated in A20-deficient microglia, a feature that has been linked to a disease-associated phenotype[14,15]. In agreement, A20-deficient microglia display a suppressed "homeostatic gene signature", a signature also reminiscent of microglia in neurodegenerative conditions[14,32,33]. Besides their microglia phenotype, A20[Cx3Cr1-KO] mice also demonstrate an aberrant molecular signature in its other CNS tissue macrophages, although less pronounced compared with microglia, in agreement with our recent finding that these macrophages share their origin and cellular dynamics with microglia[7].

Our studies demonstrate that layer V pyramidal neurons from microglia A20 knockout mice show a strong and significant increase in sEPSC frequency, suggesting an increase in cortical dendritic spine number. Previous studies already indicated that alterations in microglial number or activation can result in functional and structural changes in cortical circuits. Activation of microglia by LPS was shown to increase the frequency of sEPSC[18], while ablation of microglia results in a decrease in spine density with a corresponding reduction in miniature excitatory postsynaptic current (mEPSC) frequency[16]. Moreover, microglia depletion results in defects in learning-induced dendritic spine remodeling, with corresponding decreases in mEPSC frequency in layer V pyramidal neurons in S1 cortex[17]. Whether the synaptic phenotype in mice with microglia A20 deficiency impacts on the normal development of these mice is currently unclear, but our studies indicate that it may have consequences for normal memory and cognition. However, additional studies are required, especially since microglia activation and synaptic and cognitive dysfunction are symptoms of many neurological pathologies.

In pathological conditions, microglia are known to respond by redirecting their physiological functions and inducing the activation of inflammatory pathways contributing to the progression of disease[1]. Here we show that mice lacking expression of A20 in microglia are highly sensitive to the development of inflammatory CNS pathology. Mechanistically, this hyperinflammatory condition is caused by overactivation of the Nlrp3 inflammasome in A20-deficient microglia, identifying A20 as a brake on microglial Nlrp3 inflammasome activation and neuroinflammation. This observation is reminiscent of our previous studies demonstrating that A20 deletion in macrophages induces excessive NF-κB-dependent Nlrp3 inflammasome activation[27]. Mice with myeloid A20 deficiency develop a severe destructive polyarthritis[34], which

can be rescued in a Nlrp3-deficient background[27]. Interestingly, a recent genetic study identified loss-of-function mutations in the *A20/TNFAIP3* gene in humans causing early-onset autoinflammation. Peripheral blood mononuclear cells derived from these patients showed constitutive activation of the NLRP3 inflammasome resulting in increased secretion of active IL-1β and IL-18, suggesting that NLRP3 hyperactivation drives pathology in these patients[35]. In our study, we now show that A20 expression in microglia critically controls inflammasome activation and CNS inflammation in the EAE mouse model of MS. However, since also non-parenchymal CNS macrophages lack A20 expression, these may also have contributed to the increased EAE pathology. Indeed, because of their strategic position at CNS barriers, meningeal, perivascular, and choroid plexus macrophages might modulate immune cell entry and hence be involved in various neuroinflammatory processes[36]. New strategies allowing to specifically target the different CNS macrophage populations will be needed in order to identify their particular function and the role of A20 in CNS homeostasis and pathology.

Absence of A20 in microglia strongly sensitized mice to EAE, associated with severe CNS inflammation, demyelination, and tissue damage. This clinical phenotype could be reversed in conditions where inflammasome activation in microglia was prevented, identifying the Nlrp3 inflammasome in microglia central to MS pathology. A number of reports already suggested the involvement of inflammasomes in the development of MS[37]. Levels of IL-1β and IL-18 are upregulated in CSF and peripheral blood mononuclear cells of MS patients, and both cytokines have been shown to influence disease development in EAE. Expression of caspase-1 is elevated in MS plaques, and ATP, as well as uric acid, both activators of the NLRP3 inflammasome are upregulated in the CSF of MS patients[37]. Experimental studies using knockout mice have shown that the induction of EAE, T-cell priming, and trafficking into the CNS, is dependent on the NLRP3 inflammasome[30,38]. However, the cell types responsible for destructive inflammasome responses within the CNS had not

**Table 2 Clinical features of MOG$_{35-55}$-induced EAE in A20$^{FL}$ and A20$^{Cx3Cr1KO}$ littermates showing disease incidence (at least a score of 2), day of onset and mean maximal clinical score**

| Genotype | Incidence | Day of onset ($p < 0.001$) | Mean-max score ($p = 0.012$) |
|---|---|---|---|
| A20$^{FL}$ | 100% (6/6) | 16.2 ± 0.5 | 2.8 ± 0.3 |
| A20$^{Cx3Cr1-KO}$ | 100% (10/10) | 11.9 ± 0.3 | 3.9 ± 0.2 |

Results are displayed as mean ± SEM

**Fig. 5** Microglia A20 deficiency aggravates autoimmune CNS inflammation due to Nlrp3 inflammasome hyperactivation. **a, b** EAE was induced by active immunization of control A20$^{FL}$ ($n = 6$) and A20$^{Cx3Cr1-KO}$ mice ($n = 10$) with MOG peptide, and clinical disease development (**a**) and body weight (**b**) was followed over time. Each data point represents the mean ± SEM as estimated by the REML analysis. Changes in clinical score and relative body weight differ significantly (***$p < 0.001$; $F$-test) between genotypes across the time span. Data are representative of three independent experiments. **c** Representative images of spinal cord of control (A20$^{FL}$) and A20$^{Cx3Cr1-KO}$ mice 13 days post immunization using LFB, APP, CD3, and MAC-3 antibodies. Scale bars represent 200 μm (overview) and 20 μm (zoom). Representative images from at least four mice per group are displayed. **d** Assessment of immune cell (CD3, CD4, and CD8 T cell, Treg) infiltration in the CNS of control (A20$^{FL}$) and A20$^{Cx3Cr1-KO}$ mice by flow cytometry just before disease onset. Each symbol represents one mouse. Data are expressed as mean ± SEM and significant differences are determined by a Mann–Whitney $U$-statistical test (***$p < 0.001$, ****$p < 0.0001$). **e** Expression of inflammasome-associated factors in the spinal cord of TAM-injected control (A20$^{FL}$) and A20$^{Cx3Cr1-KO}$ mice 12 days post immunization. Each symbol represents one mouse. Data are expressed as the ratio of the mRNA expression normalized to endogenous housekeeping genes and expressed as mean ± SEM. Significant differences are determined by a Mann–Whitney $U$-statistical test (****$p < 0.0001$). **f, g** Active immunization of control (A20$^{FL}$, $n = 11$), A20$^{Cx3Cr1-KO}$ ($n = 10$), Nlrp3$^{KO}$ ($n = 18$), and A20$^{Cx3Cr1-KO}$-Nlrp3$^{KO}$ ($n = 8$) mice (**f**) and of control (A20$^{FL}$, $n = 28$), A20$^{Cx3Cr1-KO}$ ($n = 15$) and A20/caspase-1$^{Cx3Cr1-KO}$ ($n = 13$) mice (**g**), and clinical disease development over time. Each data point represents the mean ± SEM as estimated by the REML analysis. Changes in clinical score and relative body weight differ significantly between genotypes across the time span (*$p < 0.05$, **$p < 0.01$, ***$p < 0.001$; $F$-test). Graph represents combined data from three independent experiments

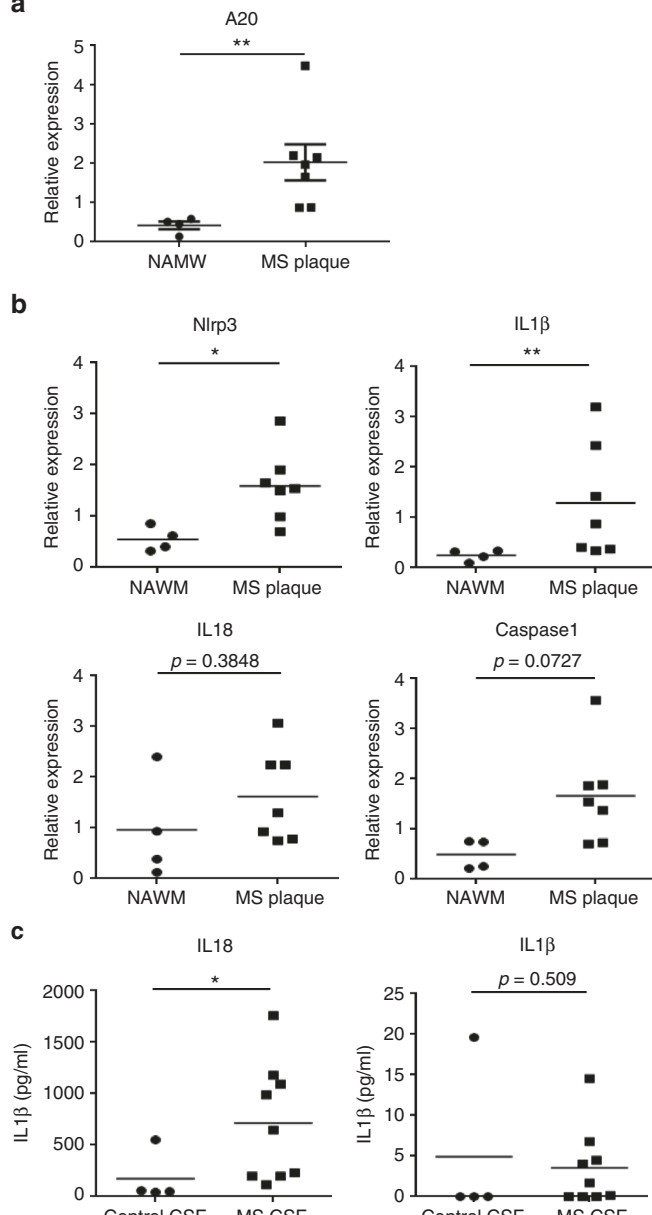

**Fig. 6** Enhanced A20 expression and NLRP3 inflammasome activation in human MS samples. **a**, **b** Relative gene expression levels of A20/*TNFAIP3* (**a**) and of inflammasome-associated factors (**b**) in plaques of postmortem MS patients (plaque) or postmortem control tissue (NAWM). Each symbol represents one patient. Data are expressed as the ratio of the mRNA expression normalized to endogenous housekeeping genes (*Sdha* and *Tbp*) and expressed as mean ± SEM. Significant differences are determined by Mann–Whitney *U*-statistical test (*p < 0.05, **p < 0.01). **c** IL-1β and IL-18 cytokine levels in the CSF of postmortem MS patients or control samples. Each symbol represents one CSF sample. Significant differences are determined by Mann–Whitney *U*-statistical test (*p < 0.05)

been investigated in MS models so far, and specifically their role in microglia was not clear. Our findings now demonstrate that a mechanism through which A20 mediates its protective effects locally in the CNS is by controlling the activation of the Nlrp3 inflammasome in microglia, thereby preventing release of pro-inflammatory cytokines IL-1ß and IL-18. This cytokine microenvironment is thought to be crucial in autoimmune CNS inflammation by potentiating the local immune response inside

the CNS[39]. Moreover, inflammatory cytokines, including IL-1ß have the potential to affect synaptic transmission, and synaptic alterations have been demonstrated in neuroinflammatory diseases including MS[40]. Since our findings demonstrate an important role for A20 in the control of inflammasome activation within microglia during EAE, this could imply a general mechanism controlling inflammatory reactions also in other neurodegenerative conditions. In this context, inflammasome activation has recently been shown to be involved in the pathology of Alzheimer disease[41], amyotrophic lateral sclerosis[42], and Parkinson's disease[43], although the specific role of microglial inflammasome activation in these neurodegenerative diseases awaits analysis.

Finally, in contrast to an important function for A20 in microglia, we could not demonstrate an important role for A20 in other neuronal cell types in the model of EAE, contrary to a previous study suggesting a role for A20 in astrocytes[44]. Although, the reasons for the different disease phenotypes are unknown, differences in the technical methodologies, mouse genetic backgrounds, and/or microbiota composition might have contributed to the differential outcomes between the studies. Spontaneous neuroinflammation characterized by reactive microgliosis, astrogliosis, and inflammatory cytokine secretion in the brain has previously been reported in A20 full knockout mice[45]. However, these mice are severely diseased and display a hyperinflammatory phenotype in nearly all tissues examined, including the brain[46]. Because of this severe lethal inflammatory phenotype, microglia-specific A20 targeting is essential to investigate the specific contribution of A20 in microglia homeostasis and pathology.

In summary, we found that A20 critically controls microglia activation both in steady-state as in conditions of neuroinflammation. In experimental models of neuroinflammation, the expression of A20 by microglia is crucial in keeping NLRP3 inflammasome activation at bay. These results may contribute to a better understanding and treatment of inflammatory and neurodegenerative diseases. Strategies targeting microglia to suppress their activation eventually through expression of A20 might prove useful for the treatment of these diseases.

## Methods

**Animals**. Conditional A20 knockout mice harboring two loxP sequences flanking exons 4 and 5 (A20$^{FL/FL}$) were generated as previously described[10]. A20$^{FL/FL}$ mice were crossed with Cx3Cr1Ert2-Cre transgenic mice[5] to generate A20$^{Cx3Cr1KO}$ mice. A20$^{FL/FL}$ mice were crossed with Nestin-Cre[47], Thy1.2-Cre[48], GFAP-Cre[49], or MOGi-Cre[50] transgenic mice to generate A20$^{CNS-KO}$, A20$^{Neur-KO}$, A20$^{ASTR-KO}$, or A20$^{ODC-KO}$ mice, respectively. Nlrp3 knockout mice have been described[51], and caspase-1$^{FL/FL}$ mice have been generated using European Conditional Mouse Mutagenesis Program ES cells[31]. R26-YFP reporter mice were obtained from The Jackson Laboratory (B6.129X1-Gt(ROSA)26Sortm1(EYFP)Cos/J). All experiments were performed on mice backcrossed into the C57BL/6 genetic background for at least six generations. Mice were housed in individually ventilated cages in either specific pathogen-free or conventional animal facilities. All experiments on mice were conducted according to institutional, national, and European animal regulations. Animal protocols were approved by the ethics committee of Ghent University.

**Histological analysis**. Mice were transcardially perfused with PBS containing 5 IU/ml heparin (De Pannemaeker, Ghent, Belgium), followed by perfusion with 4% paraformaldehyde. Brain and spinal cord tissue were dissected, dehydrated, and embedded in paraffin blocks. Alternatively, for vibratome sections brains were stored in low-melting agarose until further processing. Sections of 5 μm were stained with Luxol fast blue (Solvent Blue 38, practical grade; Sigma Genosys) for assessment of demyelination, or incubated with antibodies against CD3 (clone CD3–12; Serotex), Mac-3 (clone CD107b, M3/84; Becton Dickinson Biosciences), B220 (clone RA3-6B2; Becton Dickinson Biosciences), amyloid precursor protein (APP; clone 22C11; Millipore), Iba-1 (Wako Chemicals), or Ki67 (clone D3B5; Cell Signaling). Sections were rehydrated and incubated in antigen retrieval buffer (Dakopatts). Endogenous peroxidase activity was blocked by immersing slides in 3% H$_2$O$_2$. Nonspecific binding was blocked by incubating sections in 5% NGS and 0.1% Triton X-100. Primary antibodies were incubated overnight at 4 °C.

**Immunofluorescence microscopy, microglia quantification, and 3D reconstruction.** Fluorescence microscopy was performed using a confocal spinning disk microscope (Zeiss, Germany). Double-positive cells (Ki67[+] and Iba-1[+] positive) were counted on 6–12 representative images (267.20 × 267.73 µm) for each sample and an average was calculated. Bright-field microscopy was done using an olympus light microscope (BX51) and with a Axio Scan.Z1 (Zeiss, Germany). For 3D reconstruction of microglia, 70-µm vibratome sections from adult brain tissue were stained overnight with anti-Iba-1 (1:500) at 4 °C, followed by Dylight 594-conjugated secondary antibody (1:1000) overnight at 4 °C. Nuclei were counterstained with Hoechst. Confocal images were taken with a Leica Sp5 AOBS confocal microscope (Leica, Mannhein, Germany), using a HCX PL APO CS 40.0 × 1.25 UV oil objective. Three cells per mouse and five mice per condition were reconstructed. For live cell imaging, primary cultured microglia were seeded in an eight-well chamber (iBidi). Cell death was monitored by propidium iodide (PI) or Sytox Green uptake. Live cell imaging was performed on a Zeiss confocal spinning disk (Zeiss, Germany). Analysis was performed on Fiji, a public domain imaging software.

**RNA-seq.** RNA-seq was performed on FACS-sorted CD45[int]CD11b[+] microglia or CD45[hi]CD11b[+]CD206[+] macrophages obtained from TAM-injected A20[FL] and A20[Cx3Cr1-KO] mice. A total of 200,000 cells, stained with antibodies against CD11b, CD45, Gr1 and CD206 (eBioscience)[52] per brain were sorted in 1 ml RNAprotect cell reagent (Qiagen) and total RNA was subsequently isolated by column purification (RNeasy Plus Micro Kit, Qiagen). Generation of cDNA (Clontech SMARTer Ultra Low RNA Kit v4) and library preparation (Illumina NexteraXT custom small-sample protocol) were performed according to manufacturer's protocols. The sequencing libraries were sequenced on an Illumina HiSeq 1000 sequencer. Basecalling and Fastq file generation was performed using standard Illumina software (RTA, Casava). Fastq files were analyzed for data quality using FastQC v0.11.5[53]. FastQC: a quality control tool for high-throughput sequence data. Available online at: http://www.bioinformatics.babraham.ac.uk/projects/fastqc. Sequences were aligned to the mouse genome assembly GRCm38 as provided by the GENCODE project[54] using the Star aligner v2.5.2b[55]. Aligned reads were counted using featureCounts v1.5.1[56]. Differential gene expression analysis followed the limma/voom pipeline (limma v3.30)[57]. Pathway analysis was carried out using IPA (Qiagen). Illumina deep sequencing was performed at a genomics core facility: Center of Excellence for Fluorescent Bioanalytics (KFB, University of Regensburg, Germany). All RNA-seq datasets are accessible under GEO accession number GSE107733.

**Electrophysiology.** Mice were injected with TAM at weaning age and acute coronal slices from postnatal days 30–35 were cut on a Leica VT1200 vibratome. Mice were deeply anesthetized with isofluorane and rapidly decapitated. A length of 300-µm-thick slices were cut in a sucrose-based cutting solution (ACSF) that consisted of 87 mM NaCl, 25 mM NaHCO3, 10 mM Glucose, 75 mM Sucrose, 2.5 mM KCl, 1.25 mM NaH2PO4, 0.5 mM CaCl2, 7 mM MgCl2, 1 mM kynurenic acid, 5 mM ascorbic acid, and 3 mM pyruvic acid. Slices were allowed to recover at 34 °C for 30 min, and then maintained at room temperature in the same solution for at least 30 min before use. During recordings, slices were perfused at 1–2 mL/min with ACSF consisting of 119 mM NaCl, 2.5 mM KCl, 1 mM NaH2PO4, 11 mM glucose, 26 mM NaHCO3, 4 mM MgCl2, 4 mM CaCl2, and 0.1 mM picrotoxin (to block inhibitory currents) bubbled continuously with 95% $O_2$ and 5% $CO_2$. Layer V pyramidal neurons in somatosensory cortex were visualized by infrared differential interference (Zeiss Axio Examiner.A1). Whole-cell voltage (Vhold = −70 mV) and current clamp recordings were made. Electrode resistances ranged from 3 to 5.5 MΩ. Pipettes were pulled on a horizontal micropipette puller (Sutter P-1000) and filled with a K-gluconate-based internal solution consisting of 135 mM K-Gluconate, 4 KmM Cl, 2 mM NaCl, 10 mM HEPES, 4 mM EGTA, 4 mM Mg ATP, 0.3 mM Na GTP, adjusted to pH 7.25 and 295 mOsm. For all recordings, the number of experiments ($n$) reported in the figure legends refer to the number of cells, while ($m$) refers to the number of animals used in the experiment. sEPSCs were analyzed using the Mini Analysis program (Synaptosoft). Action potential (AP) profiling was assessed by injecting current in 50 pA steps starting at −150 mV until 850 pA. Whole-cell voltage clamp recordings were gained 5–20-fold, low-pass filtered at 1 kHz, and digitized at 3 kHz (Molecular Devices DigiData 1440A and Multiclamp 700B). Input resistance (Rin), pipette series resistance (Rs), and membrane holding current were monitored throughout all recordings to ensure stability and quality. Recordings were made at 34 °C.

**Behavioral test.** Spatial learning and cognitive flexibility was examined in the hidden platform MWM[19,20,58]. A circular pool (diameter 150 cm) was filled with tempered water (26 ± 1 °C), opacified by adding nontoxic white paint. The platform (diameter, 15 cm) was hidden 1 cm underneath the water surface. The location of the platform remained the same throughout the 10 days of spatial acquisition, was relocated to the opposite quadrant during the 5 days of reversal learning trials and removed during probe trials. Female A20[Cx3Cr1-KO] and control littermate mice were trained daily in four sequential sessions (15–30 min interval) starting randomly from one of four starting positions. Animals that were not able to find the platform within 120 s, were gently guided to the platform where they remained for

15 s before they were transferred to their home cage. Probe trials were conducted on days 6 and 11 during acquisition learning and on day 6 of reversal learning. EthoVision video tracking equipment and software (Noldus, Wageningen, The Netherlands) were used to quantify and extract swim paths parameters.

**Systemic and icv LPS injection and CSF isolation.** Female mice received an intraperitoneal injection of 3.5 mg/kg LPS (LPS-EB Ultrapure, *Escherichia coli* 0111:B4 from Invivogen) in PBS. Body temperature and survival were monitored every hour after LPS challenge. CSF was isolated from the fourth ventricle using the cisterna magna puncture method[59]. For icv injection of LPS, mice were anaesthetized with isoflurane and mounted on a stereotactic frame. A constant body temperature of 37 °C was maintained using a heating pad. Injection coordinates were measured from the bregma (anteroposterior 0.07, mediolateral 0.1, dorsoventral 0.2) and lambda. These coordinates were determined using the Franklin and Paxinos mouse brain atlas. A Hamilton needle was used to inject 4 µl LPS (0.25 µg/µl) into the left lateral ventricle of the brain.

**Induction and assessment of EAE.** Eight- to 15-week-old male littermate mice were immunized subcutaneously with an emulsion of 200 µg MOG$_{35-55}$ (MEVGWYRSPFSRVVHLYRNGK) peptide in 100 µl sterile PBS and an equal volume of CFA (Sigma-Aldrich) supplemented with 10 mg/ml *Mycobacterium tuberculosis* H37RA (Becton Dickinson Bioscience). Mice also received 50 ng of pertussis toxin (Sigma-Aldrich) in 200 µl sterile PBS at the time of immunization and 48 h later. Clinical signs of disease were scored on a scale of 0–5, with 0.5 points for immediate clinical findings as follows: 0, normal; 1, weakness of tail; 2, complete loss of tail tonicity; 3, partial hind limb paralysis; 4, complete hind limb paralysis; 5, forelimb paralysis or moribund. To eliminate any diagnostic bias, mice were scored blindly.

**T-cell recall assay.** Ten days post immunization with MOG$_{35-55}$ peptide, spleens were isolated. Erythrocytes were lysed using ACK lysis buffer and splenocytes were cultured in 96-well plates at a density of $7 \times 10^5$ cells/well in DMEM supplemented with 5% FCS, L-glutamine, nonessential amino acids (NEAA) and penicillin/streptadivin antibiotics. Cells were then treated with either no, 1, 10, or 30 µg/ml MOG$_{35-55}$ peptide. After 48 h, supernatant was collected and concentrations of IL-2 and IFNγ were determined by ELISA (eBioscience, San Diego, CA, USA).

**Quantitative real-time PCR on mouse tissue.** Total RNA was isolated using TRIzol reagent (Invitrogen) and an Aurum Total RNA Isolation Mini Kit (Bio-Rad), according to manufacturer's instructions. Synthesis of cDNA was performed using an iScript Advanced cDNA synthesis kit (Bio-Rad), according to manufacturer's instructions. A total of 10 ng of cDNA was used for quantitative PCR in a total volume of 10 µl with LightCycler 480 SYBR Green I Master Mix (Roche) and specific primers, on a LightCycler 480 (Roche). Real-time PCR reactions were performed in triplicates. The following mouse-specific primers were used: ASC forward, 5′-CTTGTCAGGGGATGAACTCAAAA-3′, ASC reverse, 5′-GCCATACGACTCCAGATAGTAGC-3′; caspase-1 forward, 5′-ACAAGGCACGGGACCTATG-3′, caspase-1 reverse, 5′-TCCCAGTCAGTCCTGGAAATG-3′; Dcaf6 forward, 5′-ACCTTTTGCCCGTGTATGG AG-3′, Dcaf6 reverse, 5′-ATTGATCCAAGCTCCCACAG-3′; Gapdh forward, 5′-TGAAGCAGGCATCTGAGGG-3′, Gapdh reverse, 5′-CGAAGGTGGAAGAGTGGGAG-3′; Hprt forward, 5′-AGTGTTGGATACAGGCCAGAC-3′, Hprt reverse, 5′-CGTGATTCAAATCCCCTGAAGT-3′; Pro-IL-1β forward, 5′-TGGGCCTCAAAGGAAAGA-3′, Pro-IL-1β reverse, 3′-GGTGCTGATGTACCAGTT-3′; Nlrp3 forward, 5′-ATTACCCGCCCGAGAAAG G-3′, Nlrp3 reverse, 5′-TCGCAGCAAAGATCCACACAG-3′; Tbet forward, 5′-AGAACGCAGAGATCACTCAG-3′, Tbet reverse, 5′-GGATACTGGTTGGATAGAAGAGG-3′; IFNγ forward, 5′-GCCAAGCGGCTGACTGA-3′, IFNγ reverse, 5′-TCAGTGAAGTAAAGGTACAAGCTACAATCT-3′; IL-2 forward, 5′-GTGCCAATTCGATGATGAGTCA-3′, IL-2 reverse, 5′-GGGCTTGTTGAGATGATGCTTT-3′; GATA3 forward, 5′-GGCAGAAAGCAAAATGTTTGCT-3′, GATA3 reverse, 5′-TGAGTCTGAATGGCTTATTCA CAAAT-3′; STAT6 forward, 5′-GGTGTTAATGCTCGAATGTGATA-3′, STAT6 reverse, 5′-CACAATGTCTCTATGTTTCT GTATGTTGAG-3′; IL13 forward, 5′-TCAGCCATGAAATAACTTATTGTTTTGT-3′, IL13 reverse, 5′-CCTTGAGTGTAACAGGCCATTCT-3′; TGFβ forward, 5′-GCTGAACCAAGGAGACGGAATA-3′, TGFβ reverse, 5′-GAGTTTGTTATCTTTGCTGTCACAAGA-3′; IL23p19 forward, 5′-TACAGAGTTAGACTCAGAACCA-3′, IL23p19 reverse, 5′-TAGTAGATTCATATGTCCCGCTG-3′ RORγt forward, 5′-CCCCCTGCCCAGAAAACACT-3′, RORγt reverse, 5′-GGTAGCCCAGGACAGCACAC-3′; Foxp3 forward, 5′-GGCCCTTCTCCAGGACAGA-3′, Foxp3 reverse, 5′-GCTGATCATGGCTGGGTTGT-3′; Ip10 forward, 5′-GTCACATCAGCTGCTACTC-3′, Ip10 reverse, 5′-GTGGTTAAGTTCGTGCTTAC-3′; IL6 forward, 5′-GAGGATACCACTCCCAACAGACC-3′, IL6 reverse, 5′-AAGTGCATCATCGTTGTTCATACA-3′ CCL2 forward, 5′-GCATCTGCCCTAAGGTCTTCA-3′, CCL2 reverse, 5′-TGCTTGAGGTGGTTGTGGAA-3′; Rantes forward, 5′- CGTCAAGGAGTATTTCTACAC-3′, Rantes reverse, 5′-GGTCAGAATCAAGAAACCCT-3′; IL-1β forward, 5′-CAACCAACAAGTGATATTCTCCATG-3′, IL-1β reverse, 5′-GATCCACACTCTCCAGCTGCA-3′. For quantitative real-time PCR on human tissue, a total of 10 ng of cDNA was used with either specific Taqman probes and TaqMan Gene Expression Master Mix (Life

Technologies) or specific primers and LightCycler 480 SYBR Green I Master Mix (Roche). The following human-specific primers were used: HMBS forward, 5′-GGGTACCCACGCGAATCAC-3′; HMBS reverse, 5′-GGCAATGCGGCTGCAA-3′; SDHA forward, 5′-CCACCACTGCATCAAATTCATG-3′, SDHA reverse, 5′-TGGGAACAAGAGGGCATCTG-3′; Nlrp3 forward, 5′-GTCTCCGA-GAGTGTTGCCTC-3′, Nlrp3 reverse, 5′-GGGACTGAAGCACCTGTTGT-3′. Taqman probes for TBP (Hs00427620_m1), Caspase1 (Hs00354836_m1), IL18 (Hs01038788_m1) and IL1β (Hs01555410_m1) were from applied Biosystems.

**Western blot**. Cell and tissue extracts were prepared in E1A lysis buffer (250 mM NaCl, 50 mM Tris pH 7.4, 0.1% NP-40) containing a complete protease inhibitor cocktail (1:25) (Roche) and centrifuged for 10 min at 14.000 r.p.m. in a micro-centrifuge at 4 °C. Supernatants were denatured in laemmli buffer, separated by SDS-polyacrylamide gel electrophoresis, transferred to nitrocellulose and immunodetected with anti-A20 (Santa Cruz Biotechnology, Inc.), anti-caspase-1 (Adipogen), and anti-actin (MP Biomedicals) antibodies.

**Microglia isolation for in vitro studies**. Zero- to 3-day-old pups were used to isolate microglia. Brains were isolated in F12 Nutrient Mixture Ham medium (Gibco Life Technologies) and stripped of olfactory bulbs, cerebellum and mid-brain, and meninges were removed. Brain tissue was digested using trypsin and resuspended in DMEM medium supplemented with 10% FCS, 1% penicillin/streptadivin, glutamine, sodium pyruvat, and NEAA. Cell suspensions were incubated in tissue flasks pretreated with poly-L-lysine. After 4–7 days, astrocytes recovered and microglia are generated by addition of DMEM medium containing 25% of L929 conditioned medium. Three to 4 days later, microglia were isolated from mixed glial cell cultures by shaking at 100 r.p.m. for an hour. Microglia were resuspended in RPMI containing 25% L929 conditioned medium. To induce cre-mediated *A20* gene deletion, cells were stimulated for 3 days with 1 μM 4-hydroxytamoxifen (Sigma). The next day, microglia were either left untreated or treated with 5 μg/ml ultrapure LPS (*E. coli* 0111:B4 strain, Invivogen) for 3 h followed by 5 mM ATP (Sigma-Aldrich) for 20 min or 20 μM nigericin (Sigma-Aldrich) for 30 min or 0.5 mg/ml silica (Min-U-Sil 5, US Silica) for 6 h. For inhibition of NF-κB, microglia were preincubated for 30 min with 1.25 μM of the selective IKK2 inhibitor TPCA-1[60] (Sigma-Aldrich) before stimulation with LPS. Time-lapse imaging of cell death in primary microglia was done based on PI or Sytox Green uptake. Microglia were either left untreated or treated with 5 μg/ml ultrapure LPS for 3 h, and imaging was started upon treatment of cells with 5 mM ATP (Sigma-Aldrich).

**Preparation of tissue samples for flow cytometry**. For preparation of single cell suspensions for flow cytometry, brain, liver, lung, heart, and skin tissues were cut into small pieces. Liver, lung, and heart tissues were transferred to enzymatic solutions (liver: 1 mg/ml collagenase D (Roche, Darmstadt, Germany); lung: 0.5 mg/ml collagenase D and heart: 1 mg/ml collagenase II supplemented with +0.1 mg/ml DNase (Sigma-Aldrich, Taufkirchen, Germany) in HBSS for 30 min or 60 min at 37 °C, shaking. For skin samples, smashed tissue was digested using 1.5 mg/ml collagenase V (Life technologies, Darmstadt, Germany) for 1.5 h at 37 °C. Cell suspensions were filtered through a mesh (70 μm, restrictively) and centrifuged for 5 min 1200 r.p.m., 4 °C. Erythrocytes were removed by RBC lysis buffer (eBioscience, San Diego, USA). Brain tissue was homogenized and filtered through a 70 μm mesh. After centrifugation (10 min 1200 r.p.m., 4 °C) the pellet was resuspended in 37% Percoll (Sigma-Aldrich, Taufkirchen, Germany) followed by centrifugation; 30 min, 800×g at 4 °C without brake.

**Microglia isolation**. Mice were perfused with cold PBS containing 5 IU/ml heparin and brains were isolated in 1X HBBS with 45% glucose and HEPES. Brains were minced using a scalpel and dissociated in high-glucose DMEM medium containing collagenase A, FCS, and DNAse I at 37 °C. Cells were isolated by centrifugation at 300×g for 10 min at 4 °C, and resuspended in 5 ml 25% Percoll and covered with 3 ml of PBS. Microglial cells are present in the pellet fraction after centrifuging at 800×g for 25 min at 4 °C, and were resuspended in FACS buffer (0.5% BSA, 2 mM EDTA in PBS) for further analysis.

**Kupffer cell isolation**. Livers were flushed with PBS, isolated and chopped into small pieces. A volume of 3 ml RPMI supplemented with 1 mg/ml collagenase A (Sigma) and 10 U/ml Dnase (Roche) was added and cells were subjected to the GentleMACS liver 1 protocol (Miltenyi Biotec). Cells were then incubated for 20 min at 37 °C in a shaking water bath and subsequently subjected to GentleMACs liver protocol 2 (Miltenyi Biotec). Suspensions were filtered, red blood cells lysed, and remaining cells were stained.

**Flow cytometry and FACS sorting**. The following antibodies were used for staining cells: anti-CD45 (clone 30-F11, eBioscience), anti-CD11b (clone M1/70, eBioscience), anti-CD64 (clone X54-5/7.1, Biolegend, Fell, Germany), anti-Gr1 (clone RB6-8C5, Biolegend), anti-F4/80 (clone BM8, eBioscience), anti-CD11b-APC-Cy7 (clone M1/70, RUO Becton Dickinson Biosciences), CD11b-BV605 (Clone M1/70, Becton Dickinson Biosciences), CD3-PECy7 (Clone 17A2,

Biolegend) CD4-eFluor450 (Clone RM4-5, eBioscience), CD8-PECy5 (Clone 53-6.7, Becton Dickinson Biosciences), Foxp3-APC (Clone, FJK-16s, eBioscience), MHCI-FITC (Clone 28-1 4-8, eBioscience), L/D-eFluor 780 (eBioscience), L/D-eFluor506 (eBioscience), and Fc receptor blocking antibody CD16/CD32 (clone 2.4G2, BD Biosciences). Prior to measuring, counting beads (Life Technologies) were added to the cells. Measurements were performed on a BD LSR II or FACSCanto II cytometer (BD Biosciences), and data were analyzed using FlowJo software (Tree Star). Kupffer cell suspensions were stained using CD45-BV510, F4/80-Biotin, CD26-AF488 (BD Bioscience), Fixable Live/Dead dye eFluor 780 (eBioscience), Streptavidin-BV605 (Biolegend), Clec4F (R&D systems), and anti-goat-AF647 (Invitrogen) for 45 min at 4 °C in the dark in a two-step staining as previously described[61]. Cells were FACS-purified with an Aria II and an Aria III (BD Bioscience) and analyzed with FlowJo software (Tree Star). Kupffer cells were gated as single live CD45+F4/80+Clec4F+ cells, and sorted to >73% purity (of total single cells in post-sort cell suspension).

**BMDM differentation and stimulation**. BMDMs were obtained from bone marrow cells flushed from mouse femurs and tibia with ice-cold sterile RPMI 1640 medium supplemented with 40 ng/ml recombinant mouse M-CSF, 10% FCS, 1% penicillin/streptadivin, glutamine, sodium pyruvate, and NEAA. Fresh M-CSF was added on day 3 and medium was refreshed on day 5. On day 7, cells were seeded and were either left untreated or treated with 5 μg/ml ultrapure LPS the next day (*E. coli* 0111:B4 strain, Invivogen) for 1 h for assessment of A20 expression.

**Peritoneal macrophage isolation**. Mice were intraperitoneally injected with 3 ml of 3% thioglycollate medium. Four days after injection, peritoneal cells were collected. Mice were anesthetized using CO₂ and injected with 5 ml PBS containing 5 mM EDTA in the peritoneal cavity. Cells were isolated from the peritoneum and centrifuged at 300×g for 3 min after which the pellet was resuspended in 1 ml of RPMI medium supplemented with 10% FCS, 1% penicillin/streptadivin, glutamine, sodium pyruvat, and NEAA. Cells were seeded and left overnight at 37 °C with 5% CO₂. Peritoneal macrophages were used the day after and either left untreated or treated with 5 μg/ml ultrapure LPS for 1 h for assessment of A20 expression.

**Bioplex and IL-1β ELISA**. Cytokine levels in culture medium and CSF were determined by magnetic bead-based multiplex assay using Luminex technology (Bio-Rad) and IL-1β ELISA (Affymetrix eBioscience), according to the manufacturers' instructions.

**Human patient samples**. Tissue and CSF specimen were obtained from the Human Brain and Spinal Fluid Resource Center (VA West Los Angeles Healthcare Center, Los Angeles), sponsored by the NINDS/NIMH, National Multiple Sclerosis Society, and Department of Veterans Affairs.

**Statistical analysis**. Results are expressed as mean ± SEM, excluding data from in vitro pooled microglia that are expressed as mean ± SD. Statistical analysis between two groups was assessed using a nonparametric Mann–Whitney $U$-statistical test. mRNA expression levels of RNA-seq data were analyzed using an unpaired $t$-test. Statistical analysis between multiple groups was assessed using either one- or two-way ANOVA with Tukey correction for multiple comparison. Relative body weight, clinical EAE scores, rectal body temperatures, and cell death assay were analyzed as repeated measurements using the residual maximum likelihood approach as implemented in Genstat v18[62]. Briefly, a linear mixed model with genotype, time, and genotype × time interaction as fixed terms, replicate as random term, and subject.time used as residual term, was fitted to data. Times of measurement were set at equal intervals and an autoregressive correlation structure of order 1 with unequal variances (i.e., heterogeneity across time) was selected as best model fit in all cases, based on the Aikake information coefficient. Significances of the fixed terms and significances of the changes in differences between genotype effects over time were assessed by an $F$-test.

**Data availability**. All RNA-seq datasets are accessible under GEO accession number GSE107733.

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

## Acknowledgements

We thank Korneel Barbry and Laetitia Bellen for animal care. We thank Keimpe Wierda for assistance with design of electrophysiology experiments and data analysis. S.V. is supported by an IWT PhD fellowship, C.M.G. and L.V.W. by FWO postdoctoral fellowships and A.S. by an FWO PhD fellowship. Research in the G.v.L. lab is supported by

research grants from the FWO, the "Geneeskundige Stichting Koningin Elisabeth" (GSKE), the CBC Banque Prize, the Charcot Foundation, the "Belgian Foundation against Cancer," "Kom op tegen Kanker," the "Interuniversity Attraction Poles program" (IAP7), and the "Concerted Research Actions" (GOA), and "Group-ID MRP" of the Ghent University. Research in the Lamkanfi lab is supported by the European Research Council grants 281600 and 683144, and the Baillet Latour Medical Research grant. M.P. is supported by the BMBF-funded competence network of multiple sclerosis (KKNMS), the Sobek-Stiftung and the DFG (SFB 992, SFB1140, SFB/TRR167, Reinhart-Koselleck-Grant) and the Sonderlinie Hochschulmedizin, project "neuroinflammation in neuro-degeneration." Research in the J.d.W. lab is supported by an FWO Odysseus Grant and European Research Council starting grant 311083.

## Author contributions

S.V., C.M.G., N.H., A.M., A.S., P.W., C.D., L.V.W., M.J.C.J., M.S., H.-K.V., D.D., G.V.I., C.L.S., and G.v.L. performed the experiments. S.V., C.M.G., N.H., A.S., P.W., C.D., O.S., E.H., A.G., M.G., S.L., C.L., R.E.V., Z.C.-V., P.C., J.d.W., M.L., M.P., and G.v.L. analyzed the data. S.J. and K.W.K. provided Cx3Cr1ERT2Cre mice. S.V., C.M.G., M.L., M.P. and G.v.L. provided ideas and coordinated the project. S.V. and G.v.L. wrote the manuscript.

## Additional information

**Competing interests:** The authors declare no competing interests.

