## [Peer Review File · Nature Communications]

Editorial Note: this manuscript has been previously reviewed at another journal that is not operating a transparent peer review scheme. This document only contains reviewer comments and rebuttal letters for versions considered at *Nature Communications*.

REVIEWERS' COMMENTS:

Reviewer #1 (Remarks to the Author):

The research article by Voet al., is an interesting paper on the novel role of microglial-derived A20 in inflammation in the CNS. The authors characterized the phenotype of microglia-specific A20 deletion, and found that it causes a hyperactivation of the basal activity in somatosensory pyramidal neurons. They also find that these mice are hypersensitive to LPS and inflammasome activation. Finally, microglial A20 deficiency worsens disease severity in EAE, which is ameliorated in A20Cx3cr1-KO nlrp-/- and caspase-1/A20Cx3cr1-KO double mutants. Finally, A20 is increased in human MS plaques as determined by qPCR.

In the revised manuscript, the authors added a considerable amount of new data, including RNAseq data from A20Cx3cr1-KO microglia demonstrating downregulation of homeostatic genes and upregulation of inflammatory signalling and disease associated microglial genes, as well as behavioural testing, which demonstrated impaired reversal learning on the Morris Water Maze test.

The majority of my comments have been addressed. Few remaining points remain:

1) In the major comment 5 I asked the authors to analyse microglial expression profile and morphological profile in A20Cx3cr1-KO nlrp-/- and caspase-1/A20Cx3cr1-KO double mutants. While the morphological analysis was not performed, in the revision the authors do analyze expression of inflammatory genes in A20 caspase double knockouts by qPCR (supplementary figure 12b). However, surprisingly, all the inflammatory genes tested are still upregulated in double knockouts (while one would expect them to be lower if A20 effect is mediated fully through caspase, as amelioration of EAE phenotype would suggest). This is a surprising result, as it suggests that there are other downstream effectors of A20 other than caspase mediating its effect on inflammatory gene upregulation. The authors need to acknowledge and discuss this surprising finding.

Minor comment:

There is no label of A20/TNFAIP3 in Figure 6a.

Reviewer #3 (Remarks to the Author):

This is the revised version of a manuscript on the role of A20 for the control of microglia activation and neuroinflammation. In their manuscript the authors make use of a multitude of techniques that involve both in vivo studies in mice and brain slice as well as primary microglia cell cultures. The authors have created an elegant but complex transgenic mouse model, in which microglia show a specific deletion of A20. They also generated

corresponding knock-out mice for astrocytes, neurons and oligodendrocytes to control for the microglia specificity of their results. As part of the revision the authors have furthermore included data on the Cx3cr1CreER: Rosa26-fl-STOP-fl-YFP reporter line to evaluate other tissue-resident macrophage populations. Overall, the data convincingly demonstrate that lack of A20 leads to an increase in the number of microglia, which in turn show an altered phenotype with an inflammatory and disease-associated signature. Along these lines, injection of LPS or induction of EAE caused a hyperinflammatory condition in A20-deficient mice that was accompanied by overactivation of the Nlrp3 inflammasome. In addition, A20 deletion was associated with defective cognitive function.

The manuscript has already undergone extensive review by three reviewers. All reviewers made very valid and constructive comments. The authors have addressed all of these comments in detail either by performing various additional experiments or by extending their discussion accordingly. I have no further comments and I strongly believe that the manuscript presents important work that should be published in Nature Communications.

Signature:

Prof. Dr. med. Stefanie Kürten

Response to reviewer's comments

We would like to sincerely thank the reviewers for their instructive and considerate comments.

Responses to the remaining comments of **Reviewer #1**

1) *“In the major comment 5 I asked the authors to analyse microglial expression profile and morphological profile in A20Cx3cr1-KO nlrp-/- and caspase-1/A20Cx3cr1-KO double mutants. While the morphological analysis was not performed, in the revision the authors do analyze expression of inflammatory genes in A20 caspase double knockouts by qPCR (supplementary figure 12b). However, surprisingly, all the inflammatory genes tested are still upregulated in double knockouts (while one would expect them to be lower if A20 effect is mediated fully through caspase, as amelioration of EAE phenotype would suggest). This is a surprising result, as it suggests that there are other downstream effectors of A20 other than caspase mediating its effect on inflammatory gene upregulation. The authors need to acknowledge and discuss this surprising finding.”*

We now included a sentence referring to this observation.

2) *“Minor comment: There is no label of A20/TNFAIP3 in Figure 6a.”*

We now corrected this and included a label in Figure 6a.